# Long-Term Changes in Corneal Endothelial Cell Density after Ex-PRESS Implantation: A Contralateral Eye Study

**DOI:** 10.3390/jcm11195555

**Published:** 2022-09-22

**Authors:** Xiaotong Ren, Jie Wang, Xuemin Li, Lingling Wu

**Affiliations:** 1Department of Ophthalmology, Peking University Third Hospital, Beijing 100191, China; 2Huangshan City People’s Hospital, Huangshan 245000, China

**Keywords:** corneal endothelial cell density (CECD), contralateral eyes, Ex-PRESS, glaucoma, intraocular pressure (IOP), long-term

## Abstract

Our purpose is to evaluate long-term changes in corneal endothelial cells after Ex-PRESS shunt implantation for the treatment of glaucoma in Chinese patients by comparison with the contralateral eye. In this retrospective observational study, glaucoma patients with a single eye undergoing Ex-PRESS shunt implantation surgery were consecutively enrolled. For each patient, the clinical assessment, including corneal endothelial cell density (CECD) before surgery and at 6, 12 months, and at last follow-up (2.43 ± 0.63 years) after surgery was reviewed. The operated eyes were in the study group and the unoperated contralateral eyes were used as the control group to compare the CECD change. A total of 48 subjects (age, 51.02 ± 17.96 years) were included. The follow-up period was 2.08~3.17 years, with an average of 2.43 ± 0.63 years. At the last follow-up after the surgery, the CECD decrease in the operated eyes (5.0%) was similar to that in the contralateral eyes (3.2%) (*p* = 0.130). There were no significant differences in CECD reduction between the two groups at baseline and each postoperative follow-up (6 months, 12 months and at the last follow-up) (all *p* > 0.05). The average IOP reduction after the surgery was 50.8%, and the number of IOP-lowering medications was significantly reduced (*p* < 0.05). In addition, visual acuity showed no significant differences during follow-up (*p* > 0.05). In this study, we found that the CECD reduction of Ex-PRESS shunt-implanted Chinese eyes was similar to that of contralateral eyes without surgery.

## 1. Introduction

Glaucoma is a group of heterogeneous diseases characterized by the gradual loss of retinal ganglion cells, cupping of the optic disc, and thinning of the retinal nerve fiber layer [1]. Epidemiological research data have shown that glaucoma is the main cause of irreversible vision loss worldwide [2]. Glaucoma has a worldwide prevalence of 3.5% of the population aged over 40 years, and 2.6% in China [2,3].

From a pathophysiological and therapeutic point of view, reducing IOP to reach the target IOP is the only evidence-based treatment, which can be achieved by the combination with drug, laser therapy, and surgical procedures [1]. Glaucoma filtration surgery is performed to reduce IOP and prevent further optic nerve damage or deterioration of the visual field when maximum tolerated drug and laser therapy does not adequately reduce IOP.

Filtration surgery is the most frequently used technique for surgical glaucoma treatment. At present, trabeculectomy is still the main operation [4]. In terms of safety, although significant progress has been made in trabeculectomy, complications such as hypotony, choroidal detachment, and anterior chamber collapse are still possible and problematic [5]. In addition, the loss of corneal endothelial cells has also gradually received attention. Corneal decompensation was reported as a late postoperative complication after trabeculectomy [6,7,8]. Furthermore, some shunt implantation surgery such as Baerveldt tube or Ahmed glaucoma valve implantation also has been paid attention to for the corneal endothelium loss [6]. Corneal complication rates of 8–29% after aqueous shunt implantation were reported [6]. 

In recent years, some new microinvasive shunting devices as an alternative to trabeculectomy have been offered, such as Ex-PRESS implant, Xen Gel stent. The damage to corneal endothelium of these implants has always been concerned, even though Xen Gel stent caused little endothelial cell damage in some studies [9,10]. The Ex-PRESS shunt is a non-valved stainless steel tube that is inserted under a partial-thickness scleral flap to connect the anterior chamber (AC) to the subconjunctival space [11,12], which is prevalent worldwide and was approved in Japan in December 2011 [13], and used widely in China since 2012. Compared with trabeculectomy, it has the advantages of reduced trauma, simple operation, and improved safety. Most studies proved that the Ex-PRESS implantation had an equivalent curative effect to trabeculectomy [14,15,16,17,18,19,20], even with more complete success [21,22]; however, some literature argued that it was less effective than trabeculectomy [23,24,25]. Regarding corneal endothelial cells, some studies have proven their safety [12,13,26,27,28]. Previous studies reported no significantly faster loss in endothelial cell counts after Ex-PRESS implantation compared with trabeculectomy [14,17,29]. While other studies reported that corneal endothelial cell density (CECD) was significantly decreased after Ex-PRESS implantation [29,30,31,32,33]. However, these studies mainly done on the Japanese [28,29,30] and some on Korean [14], Italian [34] and American [18] populations. In addition, few studies reported longer assessments for CECD loss after Ex-PRESS implantation and no study compared with the contralateral eyes. Therefore, the aim of this study was to evaluate long-term changes in corneal endothelial cells after Ex-PRESS shunt implantation in Chinese patients by comparison with the contralateral eye. 

## 2. Patients and Methods

### 2.1. Patients

In this retrospective observational study, patients with glaucoma and a single eye undergoing Ex-PRESS shunt implantation treatments at the Peking University Third Hospital Eye Center from October 2015 to June 2020 were included. The study was approved by the Ethics Committee of Peking University Third Hospital and adhered to the tenets of the Declaration of Helsinki. Unoperated contralateral eyes were used as controls to compare CECD changes.

All patients had unsatisfactory IOP control despite maximally tolerating topical medication before the surgery. We included patients who were over 18 years of age and completed regular postoperative follow-up for at least 2 years. Exclusion criteria include: (a) congenital glaucoma; (b) previous keratoplasty, preoperative corneal decompensation, corneal endothelial cell disease, and any other corneal epithelial or stromal disorders in either the operated eye or the contralateral eye; (c) patients who were performed glaucoma filtration surgery or other ocular surgery on the contralateral eye during follow-up; (d) patients with aphakic or pseudophakic eyes or other ocular disorders that would affect the corneal endothelium (including fellow eyes).

### 2.2. Surgical Procedure

All operations were performed by the same doctor (WLL, an experienced Chief Ophthalmologist). All eyes were administered retrobulbar anesthesia with lidocaine 2%. After a fornix-based conjunctival flap creation, a 0.2 mg/mL solution of mitomycin C (MMC) was applied for 3 min, followed by copious irrigation with a balanced salt solution. A rectangle scleral flap was developed, and the Ex-PRESS (model P50, Alcon Laboratories, Fort Worth, TX, USA) was introduced into the AC at the base of the scleral flap, midway between the iris periphery and the cornea. Finally, the flap was closed by 10-0 Nylon sutures combined with releasable sutures, and the conjunctiva was approximated sutured to the limbus also using 10-0 Nylon sutures. We used the balanced salt solution to reform the anterior chamber, and checked for wound leaks. A sterile eye shield was placed over the eye following the corticosteroid/antibiotic ointment. Postoperatively, topical antibiotics and steroid treatment were administered for approximately 4–6 weeks. The releasable sutures were removed within one week.

### 2.3. Preoperative and Postoperative Examinations

Age, sex, preoperative IOP, the number of IOP-lowering medications (a fixed combination agent was counted as two medications), and the diagnosis were reviewed for all patients. Apart from routine preoperative and postoperative examinations, specular microscopic examination of corneal endothelial cells was routinely performed by an experienced examiner using a noncontact specular microscope (Robo SP-8000; Konan Medical, Nishinomiya, Japan) immediately before surgery and at 6 months, 1 year and at the last follow-up (≥2 years) after surgery. This instrument automatically captures images of the endothelium once the subject fixates on a target. Then, the CECD (cells/mm^2^) and at least 50 contiguous endothelial cells centered on the screen were hand-marked, and a computer algorithm was used to calculate the values. The results on the central area of the cornea were analyzed.

The preoperative and postoperative IOPs were measured by Goldmann applanation tonometry (Haag-Streit AG, Bern, Switzerland). Types of IOP-lowering medications were recorded.

Complete success was defined as an IOP between 5 and 21 mm Hg without additional medications or/and procedures to reduce IOP, while the definition of qualified success was the same criteria mentioned above but with medications.

### 2.4. Statistical Analysis

SPSS software version 22 (SPSS Inc., Chicago, IL, USA) was used for statistical analysis. The Kolmogorov-Smirnov test was applied to determine data normality. The descriptive data are presented as the mean and standard deviation (SD). The mixed linear model was used to compare differences pre- and post-operate. To analyze repeated measurements, the eye was selected as the subject, and the time point was set as the repeated factor. For comparisons of the two groups, the paired *t*-test was used for normally continuous variables, and the Wilcoxon nonparametric test was applied for ordinal variables. *p* values less than 0.05 were considered statistically significant.

The sample size was estimated according to the standard formula for mean comparison. α and β were set as 0.05 and 0.1 respectively. Change rate of corneal endothelial cell density was selected as main outcome for sample size estimation. According to a previous study [31], it was estimated that 30 pairs of eyes were needed.

## 3. Results

A total of 48 subjects were eligible for the study. All patients were over 18 years of age, and the average age was 51.02 ± 17.96 years old. There were 36 males and 12 females. The general information of the patients is shown in Table 1. The follow-up period was 2.08~3.17 years, with an average of 2.43 ± 0.63 years.

### 3.1. Change in Clinical Parameters after Ex-PRESS Implantation

As shown in Table 2, compared with preoperative baseline, the average IOP and the number of IOP-lowering medications were significantly reduced at each follow-up time point (all *p* = 0.000). However, visual acuity (logMAR, BCVA) showed no significant differences (all *p* > 0.05).

### 3.2. Comparison of CECD in Ex-PRESS-Implanted Eyes and Contralateral Eyes

At preoperative baseline, the CECD in the operated eyes were similar to that in the contralateral eyes. Compared with preoperative baseline, no significant difference existed in CECD at postoperative 6 and 12 months either in operated eyes or in contralateral eyes (Table 3). While at the last follow-up, CECD was significantly decreased both in operated eyes and in contralateral eyes compared with preoperative CECD (*p* = 0.017, *p* = 0.012). However, no statistically significant difference in the CECD decrease percentage was found between the two groups (5.0% vs. 3.2%, *p* = 0.130). Four patients underwent bleb needling for IOP elevation. Apart from that, neither patient had apparent tube-corneal contact nor required additional intraocular surgery, including the contralateral eyes.

## 4. Discussion

This long-term follow-up study of more than 2 years showed that the CECD reduction of Ex-PRESS shunt-implanted eyes was similar to that of the contralateral eyes without surgery. To our knowledge, this was the first study to compare CECD reduction after Ex-PRESS shunt implantation between both eyes of patients. The mean CECD value is approximately 3000 cells/mm^2^ in young adults and is reduced by 0.5 ± 0.6% due to aging every year [6]. POAG patients showed a reduction of 0.68–12.3% in CECD per year (the data include multiple treatment methods) [6,35]. Our study showed that the decrease was 2.2% in shunt-implanted eyes and 1.8% in contralateral eyes at the 12-month follow-up. At the last follow-up time point (>2 years), the decreases were 5.0% and 3.2%, respectively. We speculated that it was related to glaucoma itself but had little connection with Ex-PRESS shunt implantation, even though both groups had significant CECD loss at the last follow-up time point compared with the preoperative baseline.

This study showed that at 6 and 12 months postoperatively, the reduction in CECD in the operated eyes was not significant. This was a coincidence to a short-term prospective study in Italians [34], in which none of the endothelial cell parameters changed 1 and 3 months after Ex-PRESS implantation. Our results also showed that in shunt-implanted eyes, CECD loss (5.0%) was significant at the last follow-up (>2 years). This was consistent with some Japanese studies showing a 2.5~4% mean CECD decrease 2 years after surgery, [30,31] even though these studies did not compare it with the contralateral eyes. However, other Japanese studies showed that the mean CECD decrease reached 18.0% at 2 years [31] and 10.3~30.0% at 2~5 years after Ex-PRESS implantation [32]. One of the factors influencing the reduction in CECD after Ex-PRESS surgery might be the shunt inserted position. Tojo et al. found that the trabecular meshwork insertion group had a 5.2% decrease, while the corneal insertion group had a 15.1% reduction at 2 years after implantation [36]. A reported case of partial decompensation of the corneal endothelium adjacent to the filtering bleb pointed out that the Ex-PRESS was inserted from the cornea, not the trabecular meshwork [33]. Therefore, Ex-PRESS shunt insertion into the cornea was considered a risk factor for rapid CECD loss [36]. Other than the shunt inserted position, some factors may also contribute to CECD loss after implantation, such as iris–cornea contact, contact between the corneal endothelium and the Ex-PRESS, high IOP, cytotoxicity of IOP-lowering medications, inflammation, and glaucoma itself [6,14,26,33,36,37,38]. Ex-PRESS is a medical-grade stainless steel device that has the advantage of position stability, and it has been suggested to be suitable for patients with low corneal endothelium [14]. The Ahmed valve was reported to have more CECD reduction, which maybe because of its material [34,39]. The Xen Gel Stent is a hydrophilic tube and possessed no foreign body reaction [9], which may account for less CECD loss. In our study, no patient had apparent tube-corneal contact, required additional surgery for the IOP to be excessively increased or required other intraocular surgery, which helped us to better study the natural course.

The current study found that the average IOP had a dramatic decline after Ex-PRESS implantation, even though there was a reduction of 50.8% at the last follow-up after surgery. In addition, the use of IOP-lowering medications significantly decreased postoperatively. This was similar to previous studies by others [12,26,27], yet our IOP drop seemed even higher. Perhaps patient selection, the operation of different surgeons, and the control for target IOP caused the difference. In addition, visual acuity showed no significant differences during the follow-up, which was in accord with the research by Beatriz Puerto et al. [40].

There were several limitations of the study. First, it was a retrospective study, whose sample size was relatively small, and the time of follow-up was limited. However, our research was sufficiently innovative to compare shunt-implanted eyes with contralateral eyes, which made up for it to some extent. Second, the measurement records of corneal endothelial cells did not cover the coefficient of variation (CV) or the incidence of hexagonal cells (6A). Most previous studies focused on corneal endothelial cell density count, which has been considered the most important factor. However, further study is necessary to measure CV and 6A. Third, multiple directions or o’clock positions, especially the area of the shunt, were inserted where the CECD decreased significantly faster than other areas [31]. We did not measure CECD from multiple areas, and we determined that it was more comparable to select the same position compared with the contralateral eyes. Finally, data from several types of glaucoma were combined, which might have different pathogeneses (both primary and secondary glaucoma), so it may introduce potential bias to some extent.

In summary, the CECD reduction of Ex-PRESS shunt implanted eyes was similar to that of the contralateral eyes without surgery in this long-term follow-up study, which implied that there might be no obvious connection between CECD loss and Ex-PRESS shunt implantation. Further study would benefit from an enlarged sample size, a sufficiently long period, more rigorous inclusion standards, other additional examinations, and more parameter comparisons.

## Figures and Tables

**Table 1 jcm-11-05555-t001:** Patients’ general information.

Characteristics	Number
Sex (M/F)	36/12
Age at surgery (years)	51.02 ± 17.96
Diagnosis, Number of eyes (rate)	
Primary open-angle glaucoma	42 (87.5%)
Secondary glaucoma	6 (12.5%)
pigmentary glaucoma	1 (2.1%)
angle-recession glaucoma	5 (10.4%)
Preoperative IOP (mm Hg)	27.42 ± 9.73
Number of preoperative IOP-lowering medications	2.79 ± 1.10
Preoperative corneal endothelial cell density (cells/mm^2^)	2427.69 ± 473.82
The follow up period (years)	2.43 ± 0.63
Complete success eyes (rate) *	35 (73.1%)
Qualified success eyes (rate) *	44 (91.7%)

Data are presented as numbers or means ± standard deviations. M: male; F: female; IOP = intraocular pressure. *: at the final follow-up (>2 years). Complete success was defined as IOP between 5 and 21 mm Hg without additional medications or/and procedures to reduce IOP, while the definition of qualified success was the same criteria mentioned above with or without medications. During the follow-up, 4 patients were undergone bleb needling for IOP elevation.

**Table 2 jcm-11-05555-t002:** Changes in clinical parameters after Ex-PRESS implantation.

Clinical Parameters	Baseline	6 Months	12 Months	Final Follow up (>2 Years)
N (eyes)	48	42	32	48
LogMAR BCVA	0.45 ± 0.45	0.44 ± 0.39	0.42 ± 0.41	0.39 ± 0.36
*p*		0.564	0.577	0.169
IOP (mm Hg)	27.42 ± 9.73	14.31 ± 3.43	14.16 ± 2.98	13.49 ± 2.62
*p*		0.000	0.000	0.000
medications	2.79 ± 1.10	0.67 ± 1.00	0.71 ± 0.92	0.47 ± 0.69
*p*		0.000	0.000	0.000

Data are shown as the mean ± standard deviation. logMAR = logarithm of minimum angle of resolution; BCVA = best corrected visual acuity; IOP = intraocular pressure; medications: Number of preoperative IOP-lowering medications. *p*-valve: Compared to baseline (paired *t* test with Bonferroni correction). *p* values of less than 0.05 were considered statistically significant.

**Table 3 jcm-11-05555-t003:** Comparison of CECD (cells/mm^2^) in Ex-PRESS implanted eyes and contralateral eyes.

Follow-Up Period	N	Operated Eyes	*p* ^a^	Contralateral Eyes	*p* ^b^	*p* ^c^
**Baseline**	48	2427.69 ± 473.82		2536.16 ± 435.36		0.879
**6 months**	31	2407.70 ± 593.91	0.384	2475.94 ± 292.94	0.978	0.523
**reduction (%)**		1.6 ± 12.6		1.0 ± 9.0		0.631
**12 months**	42	2364.28 ± 554.44	0.255	2417.18 ± 381.16	0.272	0.623
**reduction (%)**		2.2 ± 8.1		1.8 ± 5.0		0.572
**Final**	48	2316.70 ± 534.77	0.017	2401.47 ± 410.31	0.012	0.178
**reduction (%)**		5.0 ± 12.2		3.2 ± 7.8		0.130

Data are shown as the mean ± standard deviation. Operated eyes: Ex-Press implantation eyes; Final: final follow up (>2 years); reduction (%): Percentage decrease in CECD from baseline. *p* ^a^, *p* ^b^: Compared to baseline (paired *t* test with Bonferroni correction). *p* ^c^: Operation eyes vs. contralateral eyes (paired *t* test with Bonferroni correction). *p* values of less than 0.05 were considered statistically significant.

## Data Availability

The datasets generated and analyzed during the current study are not publicly available but are available from the corresponding author on reasonable request.

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
