# Peer review of "Long-Term Changes in Corneal Endothelial Cell Density after Ex-PRESS Implantation: A Contralateral Eye Study"

_jcm, 2022, doi:10.3390/jcm11195555_

Round 1
Reviewer 1 Report
Thank you for giving me the opportunity to review the paper of Xiaotong Ren et all tilted: "Long-term Changes in Corneal Endothelial Cell Density after 2 Ex-Press Implantation in Chinese Patients: by Contrast with Contralateral Eye".
Although the manuscript has a potential to be interesting, authors have made many typing errors which discourage the reader and suggest that it was written carelessly.
For example, in the names of authors and their affiliations:
“Xiaotong Ren,1, Jie Wang2, Xuemin Li1† and Lingling Wu 1*”
By the way: what does † mean?
There are also many errors in references section. Almost every reference in written in different style.
For example:
31. Tojo N, Hayashi A. Ex-Press® versus Baerveldt implant surgery for primary open-angle glaucoma and pseudo-exfoliation glaucoma. INT OPHTHALMOL 2021; 41(3): 1091-1101.
32. Tojo N HAMA. Corneal decompensation following filtering 327 surgery with the Ex-PRESS® mini glaucoma shunt device. Clin Ophthalmol 2015.
Also, some important reference is missing, since the CECD loss after Ex-press was not only studied in Japanese, Korean, Italian and Canadian population but also in Polish: PMCID: PMC4475547
According to the journal style abstract should be unstructured.
The number of groups is very small. Please explain how you determined the sample size for the study? Was your study powered to obtain the results in terms of change in CECD? Did you perform a sample size analysis?
Lines: 154-156: ”POAG patients showed a reduction of 11.9-31% in CECD compared with normal people. Our study showed that the decrease was 2.2% in shunt-implanted eyes and 1.8% in contralateral eyes at the 12-month follow-up. At the last follow-up time point (>2 years), the decreases were 5.0% and 3.2%, respectively.” For me is a strong discrepancy. The study groups contain patients with glaucoma, but according to the authors results the CEDC was smaller than these cited in the literature. Please explain?
The sentence in line 196:” Furthermore, few studies have compared shunt-implanted eyes with contralateral eyes” and line 151: “To our knowledge, this was the first study to compare CECD reduction after Ex-Press shunt implantation between both eyes of patients.” are contradictory.
“In summary, the CECD reduction of Ex-Press shunt implanted eyes was similar to that of the contralateral eyes without surgery in this long-term follow-up study, even though both groups had significant CECD loss at the last follow-up time point (>2 years)” (lines 208-211) – I am sorry that I conclude, that your trial is not adequately powered to arrive at your stated conclusions.
Reviewer 2 Report
The authors evaluated long term changes in CECD after ex press implantation for glaucoma treatment and they found that CECD reduction was similar to control eye without surgery.
The manuscript has an interesting topic and has a well design, but some issues should be change.
Title
In the title avoid the use of commercial names and include only general, no brand or labeled, information.
The mention of Chinese information is not relevant, the type of glaucoma could be more interesting.
And the subtitle could be improved with: A contralateral eye study
Abstract
Remove headings
Las follow up should be included in months with the mean and SD, not more than 2 years
In conclusion the term connection seems that want to express correlation?
Introduction
Include some prevalence data of the glaucoma in the world and also in the Chinese population if the manuscript only apply to Chinese population
The current treatment of the glaucoma have to be addressed in a better a manner. Include from older to the newest treatment, explain with treatment are more common and which were disappeared.
Explain the current implant similar to ex press and explain the pros and contras of all the current similar glaucoma implantation surgery devices
Prior to explain that this new device do not reduce CECD, you have to mention that the implant glaucoma surgery damaged the CECD supported with updated references
Methods
The retrospective design is a limitation and should be included in the section limitation
Why only one eye have de glaucoma implant surgery? Did the contralateral eye have glaucoma? It this case, it seems unethical to only implant one eye
Explain the inclusion and exclusion criteria with enumeration to clarify which are inclusion criteria and which other are exclusion criteria
Include a specific section for the surgery procedure and explain with more detail
Include all devices, bran, manufacturer and country of the measure instrument that were used, some of them seems to be missing in the description of the section 2.2
The criteria for the correct IOP should be reference
The medication of the subject was collected?
Include all statistical test used o the section statistical analysis
Results
Change the name of 3.1 subheading
Over 18 years should be an inclusion criteria, not in the results section
Medication data should be included in Table 1
Include race info in Table 1
Include % in diagnosis section
Describe the preoperative lowering medication
In table 2 all p value were compared with baseline? Any differences in intermediate follow up comparison? This is not clearly explained
Discussion
Is there any study in Asian population? Compare
Use the same terms to cite the ex-press implant, if possible with no brand name all label (apply to all manuscript)
Lines 185 – 191 seems a summarize of the findings that have to be included at the beginning of the discussion
In limitation section do not compare with other limitations authors research, explain your limitations only. 195 line
The not inclusion of variation and hexagonality is a flaw in the design, I imagine that no one registered this data but the instrument provided more measures.
The conclusion needs to improve in order to the scientific soundness, it appear sometimes weak
The references should be added in the journal format within all journal in lower case letters
Updated the references to after 2010 when possible
And include only manuscript research from indexed journals when possible
Round 2
Reviewer 1 Report
The answers were done quite supeficially, however most of my comments were aplied.
Reviewer 2 Report
Comments solved